# Effects of Mitochondrial Transplantation on Transcriptomics in a Polymicrobial Sepsis Model

**DOI:** 10.3390/ijms242015326

**Published:** 2023-10-18

**Authors:** Seongmin Kim, Ji Heon Noh, Min Ji Lee, Ye Jin Park, Bo Mi Kim, Yun-Seok Kim, Sangik Hwang, Chungoo Park, Kyuseok Kim

**Affiliations:** 1School of Biological Science and Technology, Chonnam National University, Gwangju 61186, Republic of Korea; sgmn0223@gmail.com; 2Department of Biochemistry, Chungnam National University, Daejeon 34134, Republic of Korea; journi@cnu.ac.kr (J.H.N.); hsi9740@naver.com (S.H.); 3Department of Emergency Medicine, CHA University School of Medicine, Seongnam 13497, Republic of Korea; minji.lee29@gmail.com (M.J.L.); yejin6577@naver.com (Y.J.P.); alfks4050@naver.com (B.M.K.); loupys@naver.com (Y.-S.K.)

**Keywords:** sepsis, immunosuppression, cytokines, rats

## Abstract

Previously, we demonstrated that mitochondrial transplantation has beneficial effects in a polymicrobial sepsis model. However, the mechanism has not been fully investigated. Mitochondria have their own genes, and genomic changes in sepsis are an important issue in terms of pathophysiology, biomarkers, and therapeutic targets. To investigate the changes in transcriptomic features after mitochondrial transplantation in a polymicrobial sepsis model, we used a rat model of fecal slurry polymicrobial sepsis. Total RNA from splenocytes of sham-operated (SHAM, *n* = 10), sepsis-induced (SEPSIS, *n* = 7), and sepsis receiving mitochondrial transplantation (SEPSIS + MT, *n* = 8) samples was extracted and we conducted a comparative transcriptome-wide analysis between three groups. We also confirmed these results with qPCR. In terms of percentage of mitochondrial mapped reads, the SEPSIS + MT group had a significantly higher mapping ratio than the others. *RT1-M2* and *Cbln2* were identified as highly expressed in SEPSIS + MT compared with SEPSIS. Using SHAM expression levels as another control variable, we further identified six genes (*Fxyd4*, *Apex2l1*, *Kctd4*, 7SK, SNORD94, and SNORA53) that were highly expressed after sepsis induction and observed that their expression levels were attenuated by mitochondrial transplantation. Changes in transcriptomic features were identified after mitochondrial transplantation in sepsis. This might provide a hint for exploring the mechanism of mitochondrial transplantation in sepsis.

## 1. Introduction

Mitochondria play vital roles in cellular metabolism, cell growth, apoptosis, calcium homeostasis, redox status, etc., and their dysfunction is being investigated as a therapeutic target for various diseases [1]. Mitochondrial transplantation was first proposed to be useful for the treatment of ischemia-reperfusion injury of the heart, and it shows encouraging clinical outcomes regarding neonatal congenital heart diseases [2,3].

Mitochondrial transplantation enhances oxygen consumption, ATP synthesis, cell viability, and inhibited oxidative stress, inflammation, and apoptosis [3,4]. Given these intriguing findings, numerous diseases are currently under scrutiny as potential candidates for mitochondrial transplantation in a range of preclinical and clinical investigations [5,6,7,8,9,10,11,12,13]. These conditions include ischemic heart disease, stroke, spinal cord injury, acute respiratory distress syndrome, inflammatory disease, etc. [4,14]. In sepsis, mitochondrial damage is critical, and therapeutic drugs to enhance mitochondrial function may be employed. The therapeutic intervention might be mitochondrial protection or acceleration of the recovery through the increase in mitochondrial biogenesis [15,16,17,18]. However, in situations of irreversible damage, mitochondrial transplantation could offer a novel and innovative strategy [10]. We previously showed that mitochondrial transplantation could have immune modulation effects in a sepsis model [13,19]. Anti-inflammatory, anti-apoptotic, and immune enhancing effects were revealed as potential mechanisms. However, more mechanisms should be investigated for the beneficial effects of mitochondrial transplantation on sepsis.

Sepsis-induced gene expression changes have been investigated, and more data suggest it would be useful in terms of investigating novel pathophysiology, biomarkers, and therapeutic targets [20,21,22,23,24,25].

Mitochondria have their own genome, and it is known that mitochondrial DNA can be transferred into recipient cells after mitochondrial transplantation [26,27,28,29].

With this background, we investigated the genetic changes after mitochondrial transplantation in sepsis. To demonstrate this, we used a polymicrobial sepsis model and observed the transcriptomic change in splenocytes (Figure 1).

## 2. Results

### 2.1. Transcriptome Analysis Based on RNA-Seq for Understanding the Effects of Mitochondrial Transplantation in Sepsis

To examine transcriptional changes in splenocytes from SEPSIS, SEPSIS + MT, and SHAM samples, we conducted a comprehensive transcriptome analysis using RNA-seq (Figure 1). An average of 51.8 million raw sequencing reads were generated from three independent experiments with a total of 25 rats (Appendix A). After trimming of the raw sequence reads, 49.9 million high-quality clean reads were mapped to the rat reference genome, with an average 96.4% uniquely mapped. To evaluate the overall transcriptome similarity among these three groups (SEPSIS, SEPSIS + MT, and SHAM), we performed principal component analysis (PCA) and found that: (1) the SHAM group was clearly separated from the others, and (2) the SEPSIS and SEPSIS + MT groups were mostly separated but not entirely (Figure 2a). These results suggested that the gene expression profiles measured in our samples are highly reproducible within multiple biological replicates for each group (10, 7, and 8 samples for SHAM, SEPSIS, and SEPSIS + MT, respectively).

We questioned whether genes of isolated exogenous mitochondria were transplanted into the samples, and thus increased mitochondrial abundance was evident. We counted the number of reads mapped to mitochondrial genome. The percentage of mitochondrial-mapped reads normalized by the total number of reads was significantly (*p* < 0.05) increased by 1.5-fold and 1.6-fold in SEPSIS + MT versus SEPSIS and SHAM, respectively (Figure 2b), indicating that our mitochondrial transplantation was technically successful.

Next, we identified 163 DEGs in the comparison between SEPSIS and SEPSIS + MT and categorized their related functions and pathways through bioinformatics analysis (Figure 3a). Three genes (*RT1-M2*, *Cyp2j10*, and *Cbln2*) had significantly higher expression levels (1.7-, 6.3-, and 4.0-folds, respectively) in SEPSIS + MT compared to SEPSIS (Figure 3b). *RT1-M2* expression followed a down–up pattern in the order of SHAM, SEPSIS, and SEPSIS + MT groups. No clear expression changes were observed for *Cyp2j10* and *Cbln2* between SHAM and SEPSIS. Additionally, we found 160 genes significantly downregulated in SEPSIS + MT. Their overall expression levels were significantly increased by 6.4-fold in SHAM and 8.1-fold in SEPSIS, but no significant difference between SHAM and SEPSIS (Figure 3c). They showed enrichment functionalities related to chromatin silencing and nucleosome positioning (Appendix A).

### 2.2. Altered Gene Expression Patterns in Response to Mitochondrial Transplantation

To further explore the detailed expression patterns of 163 DEGs, we divided them into two main groups using K-means clustering analysis. Among the order of SHAM, SEPSIS, and SEPSIS + MT, genes in the first cluster showed a decreasing expression trend (Appendix A). Intriguingly, the second cluster had genes that followed an up-and-down expression pattern (Appendix A). Among the seven genes belonging to the second cluster, five genes (*Mapt*, *Fyd4*, *Apex2l1*, *Kctd4*, and 7SK) had significantly higher expression levels (1.8-, 2.7-, 2.3-, 2.3-, and 2.5-folds, respectively) in SEPSIS versus SEPSIS + MT, and a similar increasing pattern (1.8-, 1.8-, 2.8-, 1.8-, and 3.2-folds, respectively) was observed in SEPSIS versus SHAM. However, the other two genes (SNORD94 and SNORA53) had similar expression patterns (higher expression levels at SEPSIS, lower expression level at SEPSIS + MT and SHAM), but without statistical significance (*p* > 0.05) (Figure 4). Based on our earlier bioinformatic analysis, these 10 genes (*RT1-M2*, *Cyp2j10*, *Cbln2*, *Mapt*, *Fxyd4*, *Apex2l1*, *Kctd4*, 7SK, SNORD94, SNORA53) could be potential targets of mitochondrial transplantation-induced mitoprotection in sepsis. Considering the conceptual basis of mitochondrial transplantation, we also expected that the biomarker genes involved in the post-mitochondrial transplantation might belong to the mitochondrial genome. Using DEG analysis, we compared the mitochondrial gene expression levels and found no significant difference detected amongst all pairwise comparisons (Appendix A).

### 2.3. Quantitative PCR Validation of Transcriptome Sequencing Data

We conducted quantitative PCR (qPCR) to validate the expression patterns of ten genes (*RT1-M2*, *Cyp2j10*, *Cbln2*, *Mapt*, *Fxyd4*, *Apex2l1*, *Kctd4*, 7SK, SNORD94, SNORA53). Subsequently, we compared the qPCR results with the data generated from transcriptomic analysis. Our findings confirmed that the data presented in Figure 5 and Figure 6 closely correspond to the expression patterns observed in the transcriptomic analysis for eight genes. These patterns included two downregulated and six upregulated genes in SEPSIS + MT, compared to SEPSIS.

The expression of two genes (*Cbln2* and *RT1-M2*) showed a 0.5-fold and 0.4-fold expression, respectively, in SEPSIS compared to SHAM (*p* < 0.005). In contrast, *Cyp2j10* gene showed an approximately 5.2-fold increase in expression in qPCR analysis when comparing SEPSIS to SHAM (*p* < 0.001), and it exhibited a 50% downregulation in SEPSIS + MT compared to SEPSIS (Figure 5). Additionally, six genes (*Fxyd4*, *Apex2l1*, *Kctd4*, 7SK, SNORD94, and SNORA53) showed a 2.3-fold, 4.0-fold, 7.3-fold, 12.0-fold, 3.6-fold, and 4.1-fold increase, respectively, in SEPSIS compared to SHAM, and their expression levels were lower in SEPSIS + MT than in SEPSIS. However, the *Mapt* gene appeared to be downregulated by approximately 0.3-fold and 0.2-fold, respectively, in SEPSIS and SEPSIS + MT when compared to SHAM, but this difference did not reach statistical significance (Figure 6).

## 3. Discussion

The percentage of mitochondrial mapped reads normalized by the total number of reads was significantly higher in SEPSIS + MT than in either SEPSIS or SHAM, indicating mitochondrial transplantation was associated with mitochondrial gene transplantation.

We found that in total 163 genes were differentially expressed after mitochondrial transplantation, with 160 downregulated and 3 upregulated. Out of 160 downregulated genes, 7 showed an increased expressed in sepsis, which was attenuated by mitochondrial transplantation.

*RT1-M2* expression was decreased in sepsis, which was partly reversed by mitochondrial transplantation. *RT1-M2* belongs to the major histocompatibility complex class I (MHC-I). MHC-I molecules are present on all nucleated cells, and they present peptides derived from endogenous antigens to cytotoxic T lymphocytes. During sepsis, CD8+ T cell numbers and functions are suppressed, which is associated with increased secondary infections and mortality [30]. In addition, MHC-1 can present exogenous antigens in antigen-presenting cells (APCs) like dendritic cells, macrophages, and B cells [31]. Moreover, it also regulates activation of natural killer cells [32,33]. The relationship between MHC-I and sepsis has been investigated, but the results are inconclusive. The expression of platelet MHC-1 was increased in sepsis [30]. However, other studies have shown that the expression of MHC class I molecules on airway epithelial cells can be downregulated during SARS-CoV-2 infection [34]. This downregulation may impair antigen presentation and subsequent activation of cytotoxic T cells, leading to immune dysfunction. Taken together, downregulation of MHC class I might induce immune suppression. In line with these findings, we showed that sepsis downregulated *RT1-M2*, and this downregulation was partially normalized with mitochondrial transplantation.

*Fxyd4* is elevated in sepsis, but mitochondrial transplantation decreases it. It is a part of the sodium/potassium exchanging ATPase complex, and its association with sepsis is not known. Acute kidney injury was associated with weak expression of *Fxyd4* [35], but the significance of this has not been investigated. It requires further investigation.

*Kctd4* is potassium channel tetramerization domain containing 4, which is predicted to be involved in protein homo-oliogomerization. Its association with sepsis has not been investigated.

*CBLN2*, a member of the Cerebellins (*CBLN*1-4) family, is a secreted protein that functions as a trans-synaptic cell-adhesion molecule and is predominantly expressed in motor-related brain areas [36,37]. The immune function of *CBLN2* is not known, which might be another research topic.

In validation experiments using RT-qPCR, we found that the expressional changes of not all DEGs tested matched the profiles from RNA-seq. As is already known, each analysis technique significantly differs in how it mathematically computes information about specific gene expression from raw data. Therefore, for protein-coding genes, further studies may be needed to ensure that the changes in transcripts are reflected in actual protein levels.

RNA-seq analysis identified 160 genes (DEGs) that showed a significant decrease in expression after mitochondrial transplantation in the sepsis model animals. Interestingly, 63.8 percent of genes were found to be transcripts from non-coding DNA regions whose end products are not translated into proteins. Of course, our present study does not provide any molecular details or biological relevance of the ‘mitochondria-responding’ transcripts. The sepsis-related functions of most of these transcripts have not yet been reported. Given that mitochondrial restoration significantly reduces their expression in sepsis models, our data suggest that the DEGs listed here may be potential molecular targets for future studies aimed at treating sepsis [38].

The seven transcripts, whose expression tended to increase in the sepsis group and then decrease significantly after mitochondrial transplantation, include three non-coding RNAs (ncRNAs). Among those, a small nuclear RNA (snRNA) 7SK is highly abundant (~2 × 10^5^ molecules/cell) and evolutionarily conserved. 7SK is transcribed by RNA polymerase (pol) III [39]. Earlier studies have shown that 7SK inhibits the transcription elongation factor P-TEFb during the pol II-dependent transcription [40]. It has also been reported that the majority of 7SK resides in the nucleoli, where it sequesters APOBEC3C (A3C) and perturbs the deaminase activity of A3C [41].

The other two ncRNAs, SNORD94 and SNORA53, can be categorized as small nucleolar RNAs (snoRNAs) that are RNA components in box C/D and box H/ACA snoRNPs, respectively [42]. Although they differ in the way they bind to the target ribosomal RNAs (rRNAs) or in the enzymatic activity they contribute (pseudouridylation or 2′-O-methylation), both can affect rRNA folding and ribosome function, which, in turn, contribute to gene expression and cellular maintenance [43,44].

Our recent work suggests that a nucleoli-resident ncRNA, RMRP, is expressed inside mitochondria (i.e., the matrix), albeit in relatively small amounts, and is critical for maintaining mitochondrial biogenesis by promoting organelle-specific DNA replication [45]. Interestingly, under conditions of cellular stress, steady-state levels of RMRP increase significantly, but mitochondrial levels decrease rather dramatically. Of course, the mitochondrial expression of the above three ncRNAs is still unknown, but it is quite interesting that they are all primarily found in the nucleoli, and that their expression is significantly increased by septic shock and rescued by mitochondrial complementation. Further studies are needed to determine whether these ncRNAs are involved in the crosstalk between the nucleus and mitochondria under cellular stress conditions accompanying sepsis.

In this study, many non-coding RNAs (ncRNAs) were suppressed by mitochondrial transplantation in sepsis. Long non-coding RNAs (lncRNAs) have been investigated for their involvement in a wide range of biological activities, such as genomic imprinting, chromosome modification, and transcriptional activation and repression [38]. Additionally, lncRNAs are suggested as potential therapeutic targets for sepsis-induced organ dysfunction, but there are currently limited studies [38]. With the design of this study, we could not determine the mechanism or significance of the suppressed ncRNA after mitochondrial transplantation, which requires further investigation.

In a previous study on the effects of mitochondrial transplantation in sepsis, regarding transcriptomics, it was shown that there were 146 differentially expressed genes [46]. Forty-six genes were downregulated and one hundred were upregulated, which differs from our study. The designs were different; they used the mouse cecal ligation and puncture model, and isolated mitochondria from pectoralis major muscle with a different isolation method. They used three samples from each group. We are not sure that these differences could result in different results. However, the differentially expressed genes were strongly associated with the biological processes of immune response, consistent with our study. Previously, we showed the immune modulatory effects of mitochondrial transplantation on sepsis, covering the subject in-depth through an in vivo and in vitro study [13].

In another study about heart failure, mitochondrial transplantation-induced gene networks associated with cell-cycle status, such as mitosis, cell cycle, and cell division, were the top three over-expressed GO terms [47]. In a cerebral ischemia-reperfusion model, transcriptomic data demonstrated that the differential gene enrichment pathways are associated with metabolism, especially lipid metabolism, with mitochondrial transplantation [48].

Taken together, mitochondrial transplantation might affect different gene pathways according to the disease model.

This study has several limitations. Although fecal slurry or cecal ligation and puncture models seem to be the best models for mimicking clinical sepsis [49], gaps in knowledge between preclinical and clinical sepsis complicate the translation of our findings to clinical conditions. Additionally, we could not specify the mechanisms of these changes. This requires further investigation. Finally, we could not identify which splenocytes were involved in the change in transcriptomics. There are monocytes, dendritic cells, macrophages, T and B lymphocytes, and natural killer cells in splenocytes. The next study might involve single-cell RNA sequencing. There are definite advantages in this method, but as a translational research concept, it could have limitations, since isolation of specific cells and performing qPCR would be more time-consuming than whole-cell transcriptomics.

In conclusion, we investigated transcriptomic changes after mitochondrial transplantation in sepsis, and this could be used to investigate the new mechanism of action in mitochondrial transplantation or new pathophysiology in sepsis.

## 4. Materials and Methods

### 4.1. Mitochondrial Isolation from L6 Cell Lines

Mitochondrial isolation methods were previously described [13]. We purified mitochondria from L6 (myoblast cell isolated from the skeletal muscle from a rat, ATCC; CRL-1458, Manassas, VA, USA) by differential centrifugation, which yielded mitochondrial extract as determined by bicinchoninic acid (BCA) assay using a BCA Protein Assay Kit (Thermo, Rockford, IL, USA) to confirm total protein concentration.

L6 cells were homogenized in SHE buffer (0.25 M sucrose, 20 mM HEPES, 2 mM EGTA, 10 mM KCl, 1.5 mM MgCl2, 0.1% defatted bovine serum albumin (BSA), protease inhibitor, pH 7.4) and then centrifuged at 1500× *g* for 5 min to remove cells and cell debris. After centrifugation, the mitochondria-containing supernatant was subsequently centrifuged at 20,000× *g* for 10 min. All centrifugation steps are performed at 4 °C. We tested mitochondrial function with ATP content and synthesis as in a previous study [13].

### 4.2. In Vivo Sepsis Model Induction

This study was approved by the Institutional Animal Care and Use Committee of the authors’ institute (IACUC-220052), in accordance with the National Institutes of Health Guidelines. This study was carried out in compliance with the ARRIVE guidelines. Male Sprague Dawley rats weighing 270–330 g were used. The rats were housed in a controlled environment (room temperature 20~24 °C, humidity 40~60%) with access to standard food and water ad libitum for 7 days before the experiment.

We used a body weight-adjusted polymicrobial sepsis model according to a previous study [50]. In brief, we used inhalation anesthesia with isoflurane for the short-term and then injected intramuscular Zoletil (50 mg/kg) and Xylazine (10 mg/kg) before experiments. Feces were collected from donor rats. A midline laparotomy was performed, and the cecum was extruded. A 0.5 cm incision was made in the antimesenteric surface of the cecum, and the cecum was squeezed to expel feces. The collected feces were weighed and diluted with 5% dextrose saline at a ratio of 1:3. This fecal slurry was vortexed to make a homogeneous suspension before administration into the intraperitoneal cavity. In sepsis induction, rats were anesthetized as above, and 0.5 cm midline laparotomy was performed, and fecal slurry was administered into the peritoneal cavity. The volume of cecal slurry given to each animal was adjusted on the body weight of the recipient rat. We administered subcutaneous fluid resuscitation (30 mL/kg 5% dextrose saline), and imipenem was injected subcutaneously at a dose of 25 mg/kg twice daily for 2 days. We did not use pain killers. Thereafter, the rats were randomly assigned to the 3 groups, SHAM, SEPSIS, or SEPSIS + MT. Randomization was performed by a research assistant, who was not performing the main procedure. Researchers who were performing main procedures were blinded to allocated groups. Body weight of rats could be a confounder, so we randomized with body-weight stratified method. We did not think other confounders could affect the results with this study design, so it was not controlled.

Mitochondria or DPBS were administered 1 h after sepsis induction at a dose of 200 µg via the tail vein. We euthanized the animals 24 h after sepsis induction for the collection of splenocytes.

### 4.3. RNA Extraction and Sequencing

The collected splenocytes were isolated and total RNA from each sample was extracted using Trizol reagent (Invitrogen, Carlsbad, CA, USA), and quality control and quantification were performed by a Bioanalyzer 2100 system (Agilent Technologies, Santa Clara, CA, USA) and Nanodrop ND-2000 Spectrophotometer (Thermo Scientific, Waltham, MA, USA). For RNA-sequencing (RNA-seq), the construction of libraries from total RNAs was performed using the NEBNext Ultra II Directional RNA-Seq Kit (NEW ENGLAND BioLabs, Ipswich, MA, USA) according to the manufacturer’s instructions. Briefly, the isolated mRNAs were used to ligate the adaptors, and then cDNA was synthesized using reverse-transcriptase with adaptor-specific primers. PCR was performed for library amplification, and subsequently, libraries were checked for quality, quantification, and size distribution using the TapeStation HS D1000 Screen Tape (Agilent Technologies, Santa Clara, CA, USA), and using a StepOne Real-Time PCR System (Life Technologies, Carlsbad, CA, USA). High-throughput sequencing was performed as paired-end 101 base pair reads on a NovaSeq 6000 (Illumina, San Diego, CA, USA).

### 4.4. Genome-Wide Transcriptome Analysis

All raw sequence reads were preprocessed to remove bases with low quality and adapter sequences using Trimmomatic (version 0.39) [51]. The rat reference genome (Rnor_6.0), gene model annotation files, and genome index built using hisat2-build were obtained from the Ensembl database (http://www.ensembl.org accessed on 1 March 2023) [52]. The cleaned paired-end reads were aligned to the reference genome using HISAT2 (v2.2.1) [53] with default parameters. Binary alignment map (BAM) files generated from HISAT2 aligner were further processed with StringTie (v2.2.1) and prepDE [54] to quantify transcript abundances using the transcripts per kilobase million (TPM) and counts per million mapped reads (CPM). The CPM quantification and differential expression analysis was performed using the edgeR package (v3.38.4) [55] which was subsequently employed to analyze differentially expressed genes (DEGs) with a read count > 10 in at least one sample, false discovery rate *q*-value < 0.05 and log 2 fold change > 1.

Gene clustering analysis was performed according to z-transformed TPM values of each of the gene using the K-means algorithm (implemented in the “stats” package in R software (version 4.2.1) [56] considering the Euclidean distance between them. To reduce intra-cluster variance and increase inter-cluster variance, Pearson’s correlation coefficients (*R*) between cluster centroid and data points in same cluster were calculated and excluded from further analysis if the *R* value was less than 0.85.

We performed the enrichment analysis of Gene Ontology (GO) categories and Kyoto Encyclopedia of Genes and Genomes (KEGG) pathway analysis of DEGs using DAVID (the Database for Annotation Visualization and Integrated Discovery; v2022q2 released; http://david.ncifcrf.gov, accessed on 1 March 2023) functional annotation tool [57]. The R package ggplot2 [58] was used to visualize the results.

### 4.5. Quantitative PCR (qPCR)

qPCR was performed in the CFX Connect Real-Time System (Bio-Rad, Hercules, California, USA, 788BR10054) using SYBR Master Mix (Bioneer, Daejeon, Republic of Korea, K-6253). The protocol used is as follows: pre-denaturation step at 95 °C for 5 min, followed by 40 cycles of denaturation for 5 min, 95 °C for 15 s, and annealing/extension at 60 °C for 30 s. For each sample, the delta-delta threshold cycle values were calculated as the difference between the Cq of the target gene and the Cq of the *Actb* gene. The forward and reverse primer sequences for qPCR are shown in Appendix A.

### 4.6. Statistical Analysis

The normality for the distribution of variables were tested using the Shapiro–Wilk test. A one-way ANOVA or a *t*-test was performed in the normally distributed data. If the distribution is not normal, Kruskal–Wallis test or a Wilcoxon rank sum test was applied as non-parametric test. All *p*-value < 0.05 was considered significant. All statistical analyses were performed using R software (version 3.6.2).

## Figures and Tables

**Figure 1 ijms-24-15326-f001:**
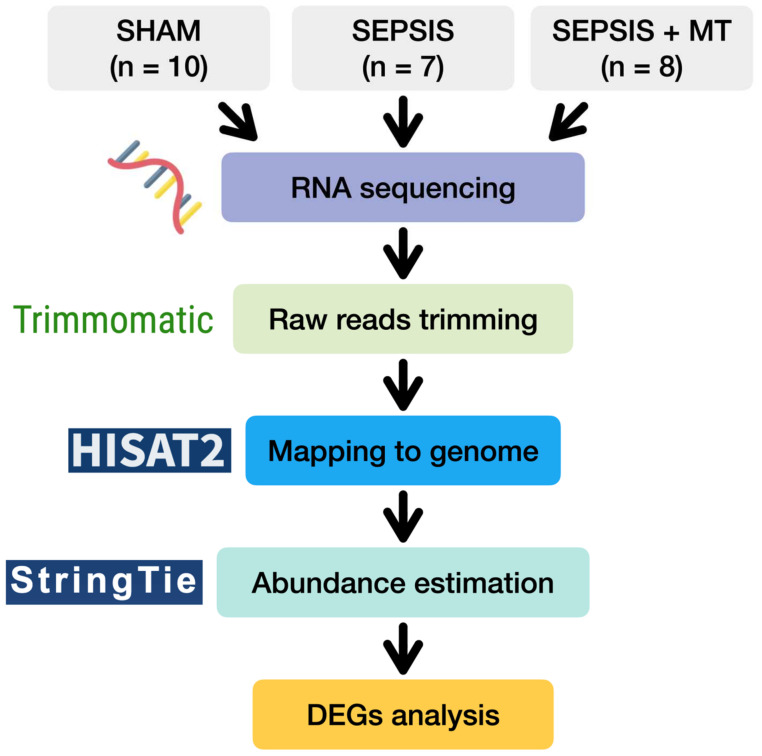
A flow diagram of transcriptome analysis of SHAM, SEPSIS, and SEPSIS + MT.

**Figure 2 ijms-24-15326-f002:**
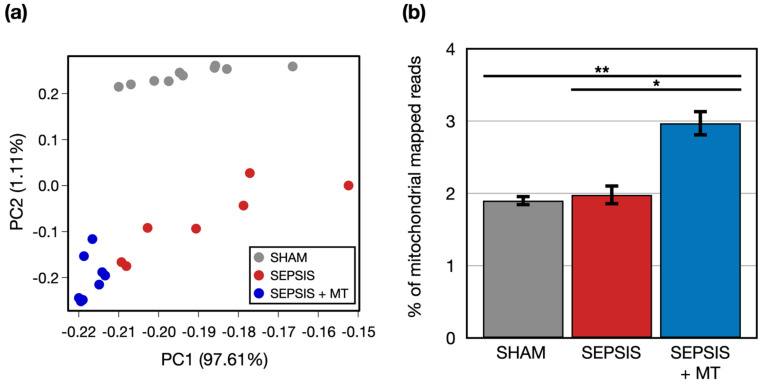
Global transcriptome profiles of three groups. (**a**) A principal component analysis (PCA) of two-dimensional data set. (**b**) A bar plot of experiments representing percentage of mitochondrial genome mapped reads. Error bar indicates standard error of mean. * and ** indicate significant difference at *p* < 5 × 10^−2^ and *p* < 5 × 10^−4^, respectively. *p* values were calculated using two sample *t*-test.

**Figure 3 ijms-24-15326-f003:**
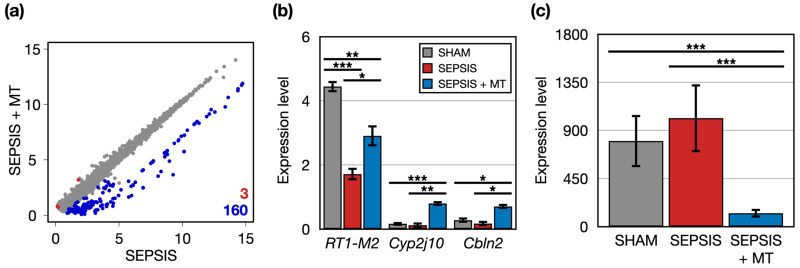
The DEGs between SEPSIS and SEPSIS + MT. (**a**) A scatter plot comparing the transcriptome profiles of SEPSIS and SEPSIS + MT. Expression level was normalized as CPM (log2-transformed counts per-million reads mapped). For each gene, the average expression level was calculated for each duplicate. Red and blue colors indicate significantly upregulated genes in SEPSIS + MT and SEPSIS, respectively. (**b**) Multiple bar plots showing gene expression levels (CPM) of three DEGs upregulated in SEPSIS + MT. *p* values were calculated using edgeR and adjusted *p* values (FDR) were shown. (**c**) A bar plot showing average expression levels (CPM) of 160 DEGs upregulated in SEPSIS. Error bar indicates standard error of mean. *, **, and *** indicate significant differences at *p* < 5 × 10^−2^, *p* < 5 × 10^−4^, and *p* < 5 × 10^−6^, respectively. *p* values were calculated using Wilcoxon rank sum test.

**Figure 4 ijms-24-15326-f004:**
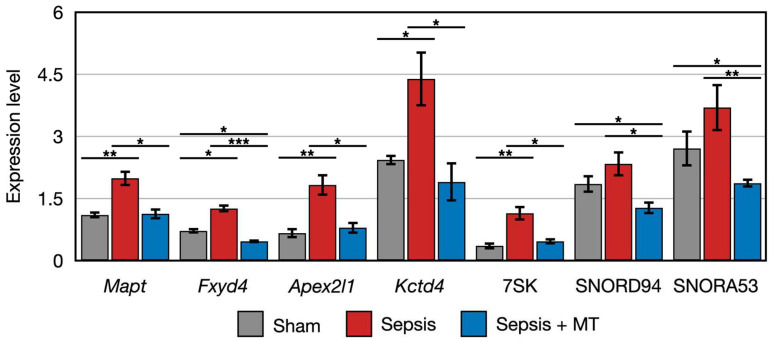
Clustered genes with similar expression pattern that were upregulated in SEPSIS compared with both SEPSIS + MT and SHAM. *, **, and *** indicate significant differences at *p* < 5 × 10^−2^, *p* < 5 × 10^−4^, and *p* < 5 × 10^−6^, respectively. *p* values were calculated using Wilcoxon rank sum test.

**Figure 5 ijms-24-15326-f005:**
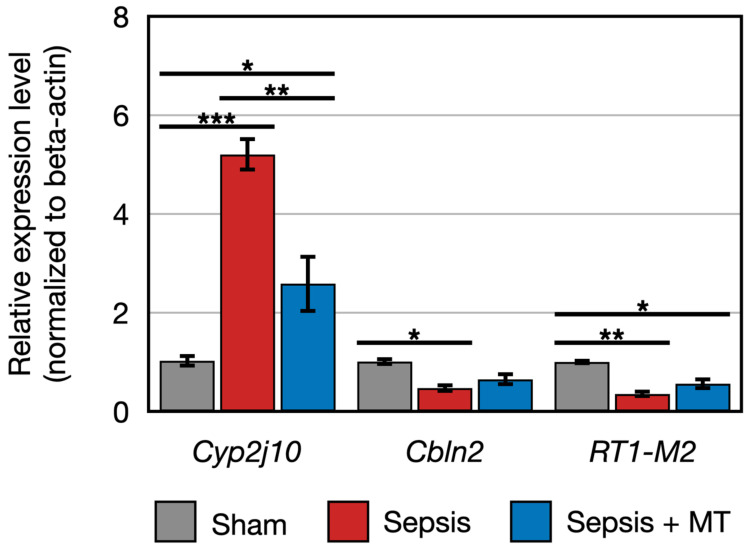
Relative expression of *RT1-M2*, *Cyp2j10*, *Cbln2* (*n* = 4 per group). Error bar indicates standard error of mean. *, **, and *** indicate significant difference at *p* < 5 × 10^−2^, *p* < 1 × 10^−2^, and *p* < 1 × 10^−3^, respectively.

**Figure 6 ijms-24-15326-f006:**
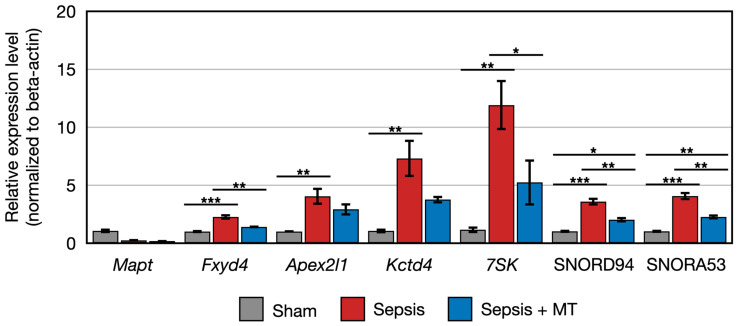
Relative expression of *Mapt*, *Fxyd4*, *Apex2l1*, *Kctd4*, 7SK, SNORD94, SNORA53 (*n* = 4 per group). Error bar indicates standard error of mean. *, **, and *** indicate significant difference at *p* < 5 × 10^−2^, *p* < 1 × 10^−2^, and *p* < 1 × 10^−3^, respectively.

## Data Availability

The datasets generated and analyzed during the current study are available from the corresponding author on reasonable request.

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
