# Peer review of "Effects of Mitochondrial Transplantation on Transcriptomics in a Polymicrobial Sepsis Model"

_ijms, 2023, doi:10.3390/ijms242015326_

Round 1
Reviewer 1 Report
The authors presented their work around the transcriptomics of sepsis and the impact of mitochondrial transplantation (MT). Briefly, using a polymicrobial sepsis model in rats, the authors investigated transcriptomics of sepsis with and without MT. Results of RNA-seq indicated potential therapeutic target genes. Given the clinical significance of sepsis, such modalities open new pathways to discover therapeutics, however, the presented work can be improved as described below.
1. Some formatting edits required:
1.1 figure captions including p values do not properly show up. An example of this is line 88, Figure 2 caption, showing 5x10-2 and 5x10-4. Superscripts need to be properly displayed, e.g. 10-2. Lines 109 and 204 show similar formatting problems in the text.
1.2 Some of the gene names were not Italicized, for example, line 142 Mapt gene is not italic, at least in the review version. Line 353 Actb gene is the same.
2. Some references are missing from the introduction:
2.1 Example 1: "Given these intriguing findings, numerous diseases are currently under scrutiny as potential candidates for mitochondrial transplantation in a range of preclinical and clinical investigations" Can some of those studies be referenced here, apart from references 4 and 5 that are used in the next sentence? It would be beneficial to specifically represent the findings concerning inflammatory conditions, including sepsis, IBD, etc.
2.2 Example 2: "In sepsis, mitochondrial damage is critical, and therapeutic drugs to enhance mitochondrial function may be employed." Can this be expanded more to elaborate on the importance of mitochondrial function? some references like "https://www.ncbi.nlm.nih.gov/pmc/articles/PMC3916385/" can be utilized.
2.3 Example 3: "we previously showed that mitochondrial 50 transplantation could have immune modulation effects in sepsis model [7,8]. However, 51 the exact mechanism is not yet known." Can previous efforts be elaborated more or mention what conclusions were made? The current work cannot be justified with "the exact mechanism is not yet known is extremely broad." One could ask why didn't authors track other mechanisms or pathways instead of mitochondria.
2.4 Example 4: in line 59, the authors mentioned using the polymicrobial sepsis model and later in the discussion pointed out the other well-known sepsis model, CLP. The use of the polymicrobial model is not justified and not well explained. It requires proper justification as to why one model was used over the other model. Do the authors suspect whether one model would change the results significantly?
2.5 Authors explained that the splenocytes were collected for this study. Can authors cite studies that show the higher relevance of splenocytes over cells sourced from other organs, such as peripheral neutrophils and platelets that are known to be key players in acute inflammatory conditions?
3. The result section was well organized. Below are feedbacks on the Results section:
3.1 Can authors provide the numerical value for the test they used to derive this conclusion "and undergo noticeable alteration during mitochondrial transplantation in the sepsis model."? (for clarity)
3.2 Authors mentioned "We first questioned whether isolated exogenous mitochondria were effectively transplanted into the samples," Can authors explain why this question came up? It can be a valid question to ask "whether the treatment was properly delivered?" However, could authors perform other experiments to test the hypothesis and establish the delivery efficiency? The question of whether the treatment was delivered through the designed pathway is still unclear and needs to be addressed.
3.3 line 121 says "these genes could be potential targets of..." Can authors clarify the reference for "these" ? The back and forth between different groups of genes can be confusing to the reader and needs to be distinguishable.
3.4 in lines 141 and 142 "and was downregulated in SEPSIS+MT compared to SEPSIS." can authors add the fold decrease of expression for more clarification?
3.5 Figures 5 and 6, n says 3-4. It is highly recommended that authors perform a complementary set of experiments to complete the data set to have n=4 uniformly across all datapoints.
3.6 Line 162 claims that " The relationship between MHC-I and sepsis has not been well investigated. " while there are a number of evidence indicating otherwise. examples listed below. It is expected that authors clarify or modify their claims based on such evidence.
3.6.1 https://pubmed.ncbi.nlm.nih.gov/33895821/
3.6.2 https://pubmed.ncbi.nlm.nih.gov/34351371/
3.6.3 https://pubmed.ncbi.nlm.nih.gov/37283946/
3.6.4 https://pubmed.ncbi.nlm.nih.gov/29126597/
3.7 In line 184, the authors claimed “The immune function of CBLN2 was not known, which might be an interesting topic.” While this line alone needs more elaboration on what makes it interesting to the authors (e.g. what are the other evidence they found? Are there any other correlations? Did their current finding alone drive such interest? Etc.), there is evidence of CBLN2 expression in the nervous system and brain tissue. That needs to be clarified unless authors would like to propose a pathway connecting the nervous system and the immune system (given minimal CBLN2 expression in naïve B cells), in which case there needs to be additional data.
3.8 In line 193 authors mentioned, “Interestingly, the majority of these were found to be transcripts from non-coding DNA regions.” Can authors quantify the “majority” and clarify what % of total transcripts?
3.9 It is suggested to remove “Nowadays, single-cell RNA seq is widely investigated.” From line 259.
The quality of English is proper and might need minor modifications.
Author Response
Dear Reviewer
I really appreciate your detail and insightful review.
As your recommendation, we revised our manuscript and it was attached.
Sincerely yours
Dr Kim

Reviewer 2 Report
Using bulk RNA-Seq, the authors investigated transcriptomic changes after mitochondrial transplantation (MT) in polymicrobial sepsis model. They identified 163 differentially expressed genes in the comparison between sepsis and sepsis+MT, and categorized their related functions and pathways through bioinformatics analysis. Most of the mitochondrial genes are downregulated, with only 3 upregulated genes. The majority of these regulated by MT in sepsis were transcripts from non-coding DNA regions, thus not many molecular details or biological relevance of these could be provided. Three genes (RT1-M2, Cyp2j10, and Cbln2) were significantly upregulated in sepsis + MT, with only RT1-M2 expression followed a down-up pattern in the order of control, sepsis, and sepsis + MT groups. Cyp2j10 and Cbln2 did not change by sepsis only but Cyp2j10 followed the up-down pattern proved by RT-PCR, together with genes Mapt, Fxyd4, Apex2l1, Kctd4, 7SK, SNORD94, SNORA53. Overall, although the MS hasn’t provided any protein level or functional study, these identified mitochondrial genes might be of interest for future mechanistic study for MT on sepsis. The written MS is acceptable, the discussion is thorough, and may be well fit for a brief report.
Minor concerns:
Line 267, what is L6 cell lines?
Housekeeping gene Actin b primer sequences are listed in the supplementary table but hasn't been shown in the figures used for the normalization of gene expression in RT-PCR.
Author Response

(The authors gave the same response as above.)
